# Vitamin D Analogs Bearing C-20 Modifications Stabilize the Agonistic Conformation of Non-Responsive Vitamin D Receptor Variants

**DOI:** 10.3390/ijms23158445

**Published:** 2022-07-30

**Authors:** Anna Y. Belorusova, Daniela Rovito, Yassmine Chebaro, Stefanie Doms, Lieve Verlinden, Annemieke Verstuyf, Daniel Metzger, Natacha Rochel, Gilles Laverny

**Affiliations:** 1Institut de Génétique et de Biologie Moléculaire et Cellulaire (IGBMC), F-67400 Illkirch, France; abelorusova@zobio.com (A.Y.B.); rovitod@igbmc.fr (D.R.); chebaro@igbmc.fr (Y.C.); metzger@igbmc.fr (D.M.); 2Centre National de la Recherche Scientifique (CNRS), UMR 7104, F-67400 Illkirch, France; 3Institut National de la Santé et de la Recherche Médicale (INSERM), UMR-S 1258, F-67400 Illkirch, France; 4Institut de Génétique et de Biologie Moléculaire et Cellulaire (IGBMC), Université de Strasbourg, INSERM U1258, CNRS UMR 7104, 67404 Illkirch, France; 5OSCAR, French Network for Rare Bone Diseases, 94270 Le Kremlin-Bicêtre, France; 6Clinical and Experimental Endocrinology, Department of Chronic Diseases and Metabolism, KU Leuven, 3000 Leuven, Belgium; stefanie.doms@kuleuven.be (S.D.); lieve.verlinden@kuleuven.be (L.V.); mieke.verstuyf@kuleuven.be (A.V.)

**Keywords:** structure function relationship, vitamin D, rare diseases

## Abstract

The Vitamin D receptor (VDR) plays a key role in calcium homeostasis, as well as in cell proliferation and differentiation. Among the large number of VDR ligands that have been developed, we have previously shown that BXL-62 and Gemini-72, two C-20-modified vitamin D analogs are highly potent VDR agonists. In this study, we show that both VDR ligands restore the transcriptional activities of VDR variants unresponsive to the natural ligand and identified in patients with rickets. The elucidated mechanisms of action underlying the activities of these C-20-modified analogs emphasize the mutual adaptation of the ligand and the VDR ligand-binding pocket.

## 1. Introduction

Following skin exposure to sunlight UV-B radiation, the secosteroid prohormone vitamin D is consecutively hydroxylated in liver and kidney to produce 1α,25-dihydroxy-vitamin D_3_ (1,25D_3_), the biologically active form. The activities of 1,25D_3_ are mediated by its binding to the Vitamin D Receptor (VDR, also termed NR1I1), a ligand-dependent transcription factor that belongs to the nuclear receptor superfamily [1,2,3]. Several loss-of-function VDR variants with impaired 1,25D_3_ binding have been identified in humans, and induce Hereditary Vitamin D-Resistant Rickets (HVDRR, OMIM 277440) [4,5,6]. The 1,25D_3_ plays a key role in calcium homeostasis by promoting VDR activities in the intestines, bones and kidneys, and is involved in a wide spectrum of biological pathways, including cell differentiation and immune responses [1,2,3]. The 1,25D_3_ binding triggers VDR conformational changes, promoting its interaction with retinoid X receptors (RXR). The VDR/RXR heterodimers bind to genomic locations, known as vitamin D response elements (VDRE), within the regulatory regions of their target genes to modulate gene transcription in concert with various co-regulatory molecules [1,2,3].

Several thousand 1,25D_3_ analogs were synthetized [7], and the crystal structures of the VDR ligand-binding domain (LBD) in the presence of 1,25D_3_ and of more than 150 analogs were solved [8,9,10]. The integrated analysis of these structures highlighted that the ligand positioning in the ligand-binding pocket (LBP) is similar, regardless of the modifications of the agonist ligands, but selective contact points between the various ligands and the 40 residues lining the LBP are formed (reviewed in [8,9]). In particular, the residues between the helices (H) 6 and 7 of the VDR LBD are described as being more flexible and positioned around the side chain of the ligand, allowing for a mutual adaptation of the ligand and the LBP.

The anti-proliferative and anti-inflammatory activities of the natural hormone are increased by inverted stereochemistry (20-epi) or the addition of a moiety at C-20 [11,12]. The crystal structures of the VDR bound to KH1060 or to MC1288, two 1,25D_3_ analogs bearing a 20-epi moiety, show that these ligands form similar interactions to 1,25D_3_. However, a few additional tighter interactions and the low energy conformation of the 20-epi analogs, which result in a higher stability of the complex, are sufficient to enhance the VDR LBP agonistic conformation [13]. In contrast, the analogs bearing a second lateral side chain at C-20 form several additional interactions with VDR that increase the size of the LBP by about 30%, resulting in a higher stabilization of the VDR-agonistic conformation [14,15]. Interestingly, among them, Gemini-72 (1α,25-dihydroxy-21(3-hydroxy-3-trifluomethyl4-trifluoro-butynyl)-26,27-hexadeutero-19-nor-20S-vitamin D3) (Figure 1A), a 1,25D_3_ analog with an additional rigid and fluorinated side chain, has potent anti-proliferative activities [16,17]. A structural analysis of VDR complexed to Gemini ligands revealed that they interact with agonist-selective amino acids, and that the highly flexible H6–H7 region of VDR adapts to dock the second lateral side chain [14,15,18,19]. In addition, Gemini-72 restores the transcriptional activity of VDRgem (L337H in zebrafish or L309H in humans), a point-mutated VDR unresponsive to 1,25D_3_ [20].

We have previously shown that BXL-62 (16-ene-20-cyclopropyl-1,25D_3_) (Figure 1A) exhibits potent transcriptional and anti-inflammatory activities [21,22]. Whereas the addition of a 16-ene modification is known to enhance the affinity for VDR by reducing the flexibility of the vitamin D side chain [23,24,25], the binding mode of a compound that presents a 16-ene and a C-20 cyclopropyl moiety was not characterized. In the present study, by solving the crystal structure of wild type (WT)- and VDRgem-bound to BXL-62, we identify the mechanisms underlying the high potency of the C-20-modified analogs to induce the transcriptional activities of VDR. In addition, we show that these modifications restore the activity of the 1,25D_3_-unresponsive VDR variants that induce HVDRR (i.e., Glu319Val, His305Gln, Trp286Arg, Leu227Pro, Ile268Thr, and Ile314Ser).

## 2. Results

### 2.1. BXL-62 and Gemini-72 Induce the Transcriptional Activity of Point-Mutated VDR Unresponsive to 1,25D_3_

To determine the VDR transcriptional activity in the presence of the C-20-derivatives of 1,25D_3_, we treated the COS cells transiently transfected with a human (h) VDR expression vector and a luciferase reporter, under the control of the promoter region of the VDR target gene, CYP24A1.

In agreement with the previous results [20], 0.1, 1, and 10 nM 1,25D_3_ for 24 h induced the luciferase activity by about 2-, 10- and 50-fold, respectively, compared to the vehicle-treated cells (Figure 1B). In contrast, Gemini-72 or BXL-62 induced it by more than 40-fold at any tested dose (Figure 1B), showing that these two C-20 modified analogs are more potent than 1,25D_3_.

Next, we investigated the potency of Gemini-72 and BXL-62 to induce the transcriptional activity of various human point-mutated VDR (Figure 2A). COS cells transiently transfected with an expression vector encoding a given hVDR variant and a luciferase reporter under the control of the *CYP24A1* promotor region were treated for 24 h with 1,25D_3_, Gemini-72, or BXL-62. The relative luciferase activity in the vehicle-treated cells transfected with the various VDR variants was at least four-times higher than in the vehicle-treated cells expressing endogenous VDR (Figure 2B), indicating that the VDR variants are overexpressed. While at 10 nM, the 1,25D_3_ had no effect, the Gemini-72 and BXL-62 fully restored the transcriptional activity of hVDR Gly319Val, His305Gln, Ile268Thr, and Ile314Ser, but not that of Trp286Arg and Leu227Pro (Figure 2B). Moreover, at 100 nM, Gemini-72 and BXL-62 were more potent than 1,25D_3_ to induce the transcriptional activities of hVDR Trp286Arg and Leu227Pro (Figure 2C,D). In addition, BXL-62 and Gemini-72 had a similar potency to induce hVDRgem transcriptional activity at all of the doses tested, whereas 1,25D_3_ had no effect (Figure 2E). These results show that the ligands bearing a C-20 modification are more potent than 1,25D_3_ to induce VDR transcriptional activity, and are sufficient to restore the activity of the VDR variants with impaired 1,25D_3_ binding.

### 2.2. Binding Mode of BXL-62

To investigate the structural modifications induced by the addition of 16-ene and C-20 cyclopropyl moieties to the 1,25D_3_ backbone, we determined the crystal structure of the zebrafish (z) VDR LBD in complex with BXL-62. The protein was crystallized in the presence of an excess of BXL-62, and of the human NCoA1 peptide encompassing the second nuclear receptor LXXLL-interacting motif. The structure was solved by molecular replacement and refined to 2.7 Å (Appendix A). BXL-62 adopted a similar orientation in the LBP as 1,25D_3_, and the BXL-62-bound VDR had the canonical VDR-agonist conformation (Figure 3A; Appendix A). Even though the 16-ene moiety reduced the flexibility of the skeleton of 1,25D_3_ [25], the interactions between VDR and the A-, Seco B-, and C/D- rings of BXL-62 were similar to those of 1,25D_3_ (Appendix A). In addition, the hydroxyl group in C1, C3, and C25 of BXL-62 and of 1,25D_3_ formed a similar interaction with zSer265 and zArg302, with zTyr175 and zSer306, and with zHis333 and zHis423, respectively (Appendix A). Moreover, all of the contacts identified between zVDR and the 1,25D_3_ lateral side chain were conserved in the zVDR-BXL-62 complex (Figure 3B; Appendix A), as well as the additional interactions with the residues of H6 (zVal328), loop (L) 6–7(zHis333), H7 (zLeu338, zLeu341), and H11 (zHis423) that were formed in the presence of BXL-62 (Figure 3C; Appendix A). These contacts were induced by the positioning of the side chain, as shown by the torsion angles of C16-C17-C20-C22 (−130° for BXL-62 and -29° for 1,25D_3_) imposed by the 16-ene configuration and the C-20 cyclopropyl group of BXL-62, resulting in the cyclopropyl pointing towards H7 (Figure 3C). In addition, the distances between the C25 geminal methyl groups and the C-terminal residues zLeu440 (loop 11–12) and zPhe448 (H12) were reduced in the presence of BXL-62 than in the presence of 1,25D_3_ (Figure 3B,C; Appendix A and Appendix A). Note that the molecular dynamics (MD) simulations indicated that the binding free energy for the ligand binding is similar for 1,25D_3_- and BXL-62-bound VDR (Appendix A). Thus, the increased rigidity imposed by the geometry of the 16-ene configuration and of the C-20 cyclopropyl group, as well as the stronger/additional interactions, contribute to a more stable VDR agonist conformation.

### 2.3. BXL-62 Stabilizes VDRgem Active Conformation

To determine whether similar structural modifications are observed in the VDR variants, BXL-62 in complex with zVDRgem (L337H) [20] was crystallized, solved, and refined to 2.4 Å (Appendix A). All of the interactions observed in the zVDR WT-BXL-62 complex were conserved in zVDRgem-BXL-62 (Figure 4A,B), notably the specific and tighter interactions involving the C-20 cyclopropyl moiety (Appendix A). The BXL-62 interacted more strongly with H337 of zVDRgem through the C-21 atom of the cyclopropyl than that of zVDR WT (Appendix A). The stronger interactions formed by the side chain of BXL-62 with residues of H6, H7, and of the C-terminus stabilized the active conformation of VDRgem, in contrast to the 1,25D_3_ that exhibited weakened hydrogen bonds between 25-OH and zHis333 and zHis423 (Appendix A).

We have previously shown that Gemini-72 forms additional contacts with zVDR WT and zVDRgem residues of H3 (zLeu255, zLeu258, and zVal262) and of C-ter (zLeu430, zLeu440, and zPhe448) than 1,25D_3_, due to its large second lateral side chain [20]. To reveal key dynamical features in the zVDR-BXL62 complex in comparison with the zVDR LBD- Gemini-72 and -1,25D3 complexes, we performed MD simulations. The most critical interactions between the zVDR residues and the analogs involved in the key hydrogen bonds (i.e., zVDR zSer265, zArg302, zSer303, zSer306, zHis333, and zHis423) were similar in the presence of the three ligands (Figure 4C). Our analysis revealed that BXL-62 and Gemini-72 exhibit similar profiles of averaged contact probabilities with VDRgem than with VDR WT (Figure 4C,D). In contrast, 1,25D_3_ failed to stabilize the interaction with the zVDRgem His423 (Figure 4C and Table 1). This different interaction between BXL-62 and 1,25D_3_ with His423 is due to the His337 having a stronger contact probability with His423 in the zVDRgem-BXL-62 complex (Table 1). The presence of the cyclopropyl moiety prevented the interaction between these two histidine residues, while showing an increased interaction with His423 (Appendix A). Taken together, these results demonstrate that the addition of the cyclopropyl to the 1,25D_3_ skeleton is sufficient to restore the VDRgem agonist conformation and dynamics.

### 2.4. In Vivo Effect of BXL-62

We have shown that the mice bearing the VDR L304H (VDRgem) exhibit rickets with a more severe phenotype than the VDR-null mice. In addition, the VDRgem mice are unresponsive to 1,25D_3_, but not to Gemini-72 [20]. As BXL-62 has higher anti-inflammatory activities than 1,25D_3_ in mice [22], and restores VDRgem transcriptional activity in vitro, we determined its potency to induce the activities of VDR variants in vivo. The WT and VDRgem mice were treated orally with either vehicle or BXL-62 for 6 h, and the duodenal transcript levels of two well-known VDR target genes (i.e., *Slc37a2* and *Slc30a10*) were determined (Figure 5A,B). In the vehicle-treated mice, duodenal *Slc37a2* and *Slc30a10* transcript levels were at-least 1.5-fold lower in VDRgem than in WT, in agreement with the previous results [20]. In addition, they were induced by at least two-fold, 6-h after administration of 3 or 10 µg/kg BXL-62 to the WT and VDRgem mice (Figure 5A,B). Next, the VDRgem were treated orally for 4 days with 10 µg/kg and the serum calcium levels were determined 24 h after the last administration. Our results show that BXL-62 increases the serum calcium levels of VDRgem mice by three-fold (Figure 5C). Thus, BXL-62 restores the VDRgem activities in mice.

## 3. Discussion

In the present study, we investigated the mechanism of action underlying BXL-62 activities and unraveled the importance of the flexibility of the VDR LBP to accommodate C-20 modified ligands.

Some analogs of 1,25D_3_ have been shown to restore the transcriptional activity of a given VDR variant unresponsive to 1,25D_3_, thus acting as VDR variant-specific drugs [5,26,27,28]. Here we show that the highly potent C-20 modified analogs, BXL-62 and Gemini-72, restore the transcriptional activities of various VDR variants affecting 1,25D_3_ binding, including the Leu227Pro and Trp286Arg that strongly impaired the 1,25D_3_-induced VDR transcriptional activity [29,30]. In addition, we show that BXL-62 is able to restore the transcriptional activity of the VDRgem variant, that was designed to respond selectively to gemini ligands and to be unresponsive to 1,25D_3_. Whereas the VDR conformational changes induced by Gemini-72 second side chain were shown to overcome the loss of activity of 1,25D_3_ on VDRgem [20], the C-20 cyclopropyl moiety of BXL-62 stabilizes the VDR LBP and enhances some of the VDR interactions, notably with zHis423. These conformational adaptations are sufficient to restore the activities of VDR variants. Remarkably, we showed that four consecutive per os administrations of BXL-62 increased serum calcium levels to a high range in VDRgem mice. However, as the dose used was 10-times higher than the BXL-62 maximum-tolerated dose identified in wild-type mice [21,22], a lower dose and/or a different regimen should normalize the calcemia, as observed for the treatment with Gemini-72 [20].

Simple inverse stereochemistry at C-20 has been shown to improve the transcriptional activity through an entropic gain of the ligand conformation, and closer contacts with His305 (zHis333) and His397 (zHis423) [13], and 16-ene modification in 1,25D_3_ increases the binding affinity for VDR [24]. Although the conformational space of BXL-62 is similar to that of 1,25D_3_ and of 20-epi analogs, a mutual adaptation of the 16-ene and C-20 moiety and the VDR LBP lead to alternate contacts. The BXL-62 side chain is rigid, due to the structural cooperation of the 16-ene and the C-20 cyclopropyl group leading to a different position of the side chain in the VDR pocket than 1,25D_3_, with the cyclopropyl group pointing towards H7 to form specific interactions with the residues of H6-H7 and with C-ter H11-H12. Importantly, the contacts between BXL-62 and H6-H7 of VDR are essential for H11 repositioning through a network involving H6 (Val328), loop 6–7 (His333), and H11 (His423). Therefore, these stronger contacts result in the stabilization of the active VDR conformation and coactivator recruitment, even though the BXL-62 and 1,25D_3_ have a similar total binding affinity for WT VDR. Of note, a similar stabilization is also observed in the presence of Gemini-72, but does not involve similar atoms. Moreover, the MD analysis of the VDR WT and of the VDRgem complexes identified His423 as a key dynamic residue involved in the stabilization of the agonistic conformation. Here, we demonstrated that, in addition to its lower CYP24A1-induced degradation due to the addition of the 16-ene, and the activity of its 24-oxo metabolite [12,21], BXL-62 stabilized the VDR complex by a combination of enthalpic (additional and tighter intermolecular contacts) and entropic (rigid side chain) effects.

## 4. Conclusions

In summary, by combining functional, structural, and MD analysis, we highlight the importance of a C-20 moiety to potentiate VDR activity, and provide, at atomic resolution, mechanistic details. The geometry imposed by the 16-ene configuration and the C-20 cyclopropyl in BXL-62 is energetically more favorable and is responsible for increased VDR interactions. Importantly, these specific properties restore the transcriptional activity of VDR variants with impaired 1,25D_3_ binding.

## 5. Materials and Methods

### 5.1. Chemicals

The 1,25D_3_ (a gift from Antonio Mourino), BXL-62, and Gemini-72 (a gift from Hubert Maehr) were dissolved in ethanol at 10^−2^ M and stored at −20 °C. The compounds were >95% pure, as determined by HPLC [21,31]. The NCoA1 (686-RHKILHRLLQEGSPS-700) peptide was synthesized at the IGBMC peptide synthesis common facility.

### 5.2. Cells

The mycoplasma-free fibroblast-like cells, derived from monkey kidney tissue (COS-7) cells (American Type Culture Collection, Manassas, VI, USA), were grown in Dulbecco’s modified Eagle’s medium (DMEM), supplemented with 10% fetal calf serum (FCS) and 40 µg/mL gentamicin. Prior to the experiments, the cells at 80% confluency were grown for 24 h in charcoal-treated FCS medium.

### 5.3. Transactivation Assays

The COS-7 cells were seeded into 24-well plates (10^5^ cells per well) and transfected with 250 ng of the expression vector encoding the full-length human (h) VDR WT, hVDRgem (Leu309His) or point-mutated hVDR (i.e., His305Gln, Trp286Arg, Leu227Pro, Ile268Thr, Ile314Ser, and Gly319Val), 500 ng of the reporter plasmid encompassing the region −414 to −64 of the human *CYP24A1* gene fused with the thymidine kinase promoter driving the firefly luciferase reporter gene, and 250 ng of the pCH110 vector encoding β-galactosidase [20] (used as an internal control) with the X-tremeGENE™ HP DNA Transfection Reagent (Merck, Darmstadt, Germany), according to the manufacturer’s instructions. After twenty-four hours, the cells were treated for twenty-four hours with indicated doses of 1,25D_3_, Gemini-72, BXL-62, or vehicle (EtOH) and the cellular lysates were assayed for luciferase activity, as recommended by the supplier (Perkin-Elmer, Waltham, MA, USA), and normalized to β-galactosidase activity determined at the optical density of 420 nm.

### 5.4. Crystallization and Structure Determination

The expression and purification of the VDR ligand-binding domain (LBD) (residues 156–453) of the zebrafish (z) wild type (zVDRwt) and the Leu337His point-mutated (zVDRgem) were performed as described [20]. The purified proteins were concentrated to 3–7 mg/mL, using Amicon ultra-30 (Merck Millipore, Molsheim, France) and incubated with a two-fold excess of BXL-62 and a three-fold excess of the coactivator NCoA1 peptide. The crystals were obtained in 50 mM Bis-Tris pH 6.5, 1.6 M lithium sulfate and 50 mM magnesium sulfate, cryo-protected with 20% glycerol, mounted in a fiber loop and flash-cooled under a nitrogen flux. The data for zVDRwt and zVDRgem complexes from a single frozen crystal were collected at 100 K on the ID29 and ID23-2 beamlines (European Synchroton Radiation Facility, Grenoble, France), respectively. The raw data were processed with XDS [32] and scaled with AIMLESS [33] programs. The structure was solved and refined using Phenix [34], BUSTER [35], and iterative model building, using COOT [36]. Crystallographic refinement statistics are presented in Appendix A.

### 5.5. Molecular Dynamics

For all of the structures, four G residues were added, using Modeler in the missing loop (residues 191 to 250 of zVDR) of the crystal structures of the complexes. Binding free energy decomposition calculations were performed on 100 ns molecular dynamics simulations of each complex and repeated five times. The first 10 ns of the simulations required for system equilibration were removed. For each complex, five independent 100 ns MD simulations were computed and averaged, and the MD trajectories at the residue level interactions were analyzed, with a distance threshold of 3.5 Å. The protein–ligands contacts were analyzed by calculating the averaged contact probabilities at the end of the averaged simulations.

### 5.6. Mice

The C57BL/6 J (WT) and VDRgem mice [20] from a similar genetic background were maintained in a temperature- and humidity-controlled animal facility and fed ad libitum (Safe diets D04, Safe, Augy, France). Ten-week-old male mice were orally administered with 100 μL of sunflower oil (Auchan, Illkirch, France), containing less than 0.01% EtOH (vehicle). The animals were killed by cervical dislocation and the tissues were immediately collected and frozen in liquid nitrogen. All of the animal experimental protocols were conducted in compliance with French and EU regulations on the use of laboratory animals for research and approved by the IGBMC Ethical Committee and the French Ministry of Higher Education Research and Innovation (#10047-2017052615101492 and #21776-2019082318288737).

### 5.7. Analysis of Transcript Levels

The total RNA was isolated using NucleoSpin kit reagents (Macherey-Nagel GmbH & Co. KG, Hoerdt, France), according to the manufacturer’s protocols. The RNA was quantified by spectrophotometry (Nanodrop, Thermo Fisher, Illkirch, France), and cDNA was prepared using 2 µg of total RNA, random hexamers, and SuperScript IV reverse transcriptase (Thermo Fisher, Illkirch, France), following the manufacturer’s instructions. Quantitative PCR (Light Cycler 480-II) was performed using the Light Cycler 480 SYBR Green I Master X2 Kit (Roche Diagnostics, Meylan, France), according to the supplier’s protocol. The data were analyzed using the standard curve method, following the manufacturer’s protocol (Lightcycler 480 II, Roche Diagnostics, Meylan, France), and the 18S housekeeping gene as internal control.

The set of primers were the following:*Slc37a2*, 5′-TAGGGCCAGACTAGAGCCA-3′ (sense) and5′-ACATGCTCATCTCTGCCGAC-3′ (antisense);*Slc30a10*, 5′-GGTGATTCCCTGAACACCGA-3′ (sense) and5′-ACGTGCAAAAGAACACCTCTG-3′ (antisense) and18 S, 5′-AGCTCACTGGCATGGCCTTC-3′ (sense) and5′-CGCCTGCTTCACCACCTTC-3′ (antisense).

### 5.8. Blood Sample Collection and Analysis

The blood was collected by inferior palpebral vein puncture, kept for 2 h at 4 °C, and centrifuged at 400× *g* for 10 min at 4 °C. The serum calcium levels were determined using colorimetric assays (MAK022, Sigma Aldrich, Saint-Quentin-Fallavier, France), according to the supplier’s protocol.

### 5.9. Data and Statistical Analysis

Data are shown as means + SD. ANOVA followed by a post-hoc *t*-test were used to evaluate the differences between the conditions. The *p* values < 0.05 (*) were considered statistically significant.

## Figures and Tables

**Figure 1 ijms-23-08445-f001:**
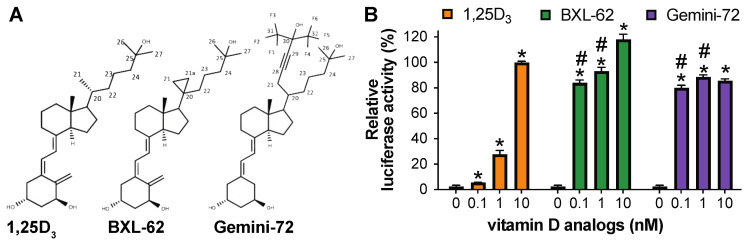
Structure and transcriptional activity of the C-20 modified analogs BXL-62 and Gemini-72. (**A**) Chemical structures of 1,25D_3_ and analogs; (**B**) Fold change of the relative activity of a reporter gene under the control of the promoter region of *CYP24A1* in COS cells transiently transfected with hVDR, and treated for 24 h, as indicated. The results are shown as % of the relative activity determined in cells overexpressing WT VDR and treated with 10 nM of 1,25D_3_. *n* = 3 biological replicates/condition. * *p* < 0.05 vs. vehicle. # *p* < 0.05 vs. a similar dose of 1,25D_3_.

**Figure 2 ijms-23-08445-f002:**
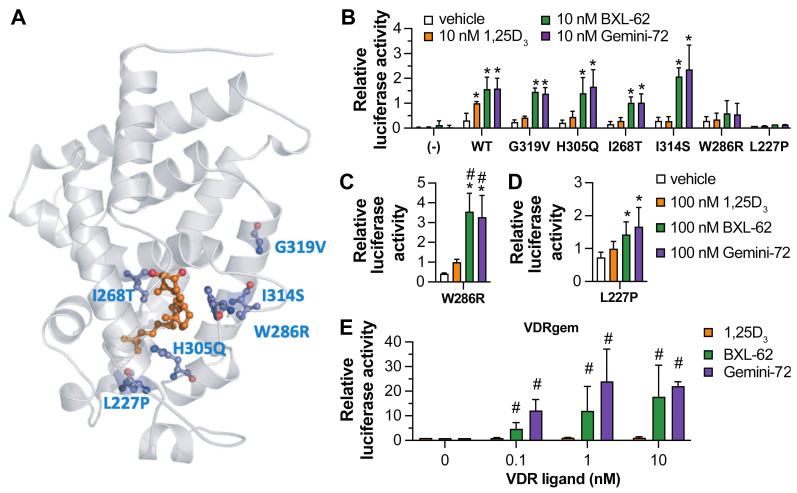
Transcriptional activity of point-mutated VDR. (**A**) VDR variants (in blue) identified in HVDRR patients mapped onto the hVDR LBD structure in the presence of 1,25D_3_ (orange) (PDB 1DB1); (**B**–**E**) Relative activity of a reporter gene under the control of the promoter region of *CYP24A1* in COS cells transiently transfected or not (−) with a VDR variant. Cells were treated for 24 h with the indicated ligand. The results are shown as fold change compared to the condition treated with 10 nM 1,25D_3_ and overexpressing WT VDR (**B**), treated with 100 nM 1,25D_3_ and overexpressing VDR W286R (**C**) or L227P (**D**), and treated with 10 nM 1,25D_3_ and overexpressing VDRgem (**E**). *n* = 3–6 biological replicates/condition. * *p* < 0.05 vs. vehicle. # *p* < 0.05 vs. 1,25D_3_.

**Figure 3 ijms-23-08445-f003:**
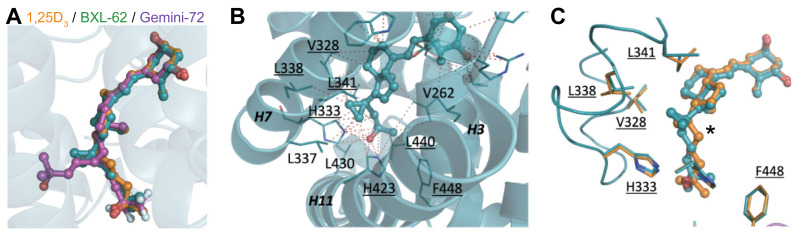
Crystal structure of zVDR—BXL-62. (**A**) Superposition of 1,25D_3_, BXL-62 and Gemini-72 ligands in the crystal structures of the zVDR LBD complexes (zVDR-1,25D_3_: PDB 2HC4; zVDR-BXL-62: PDB 7BNS; zVDR-Gemini-72: PDB 3O1D); (**B**) Close-up view of the VDR ligand-binding pocket around the BXL-62 side chain (in blue). Residues that contact the ligand with a cutoff of 4.0 Å are labelled. Residues that form additional or stronger contacts with BXL-62 compared to 1,25D_3_ are underlined; (**C**) Superposition of BXL-62 (blue) and 1,25D_3_ (orange) side chains within the VDR ligand-binding pocket with residues that form stronger contacts with BXL-62 than with 1,25D_3_ are highlighted. * indicates the torsion angles of C16-C17-C20-C22.

**Figure 4 ijms-23-08445-f004:**
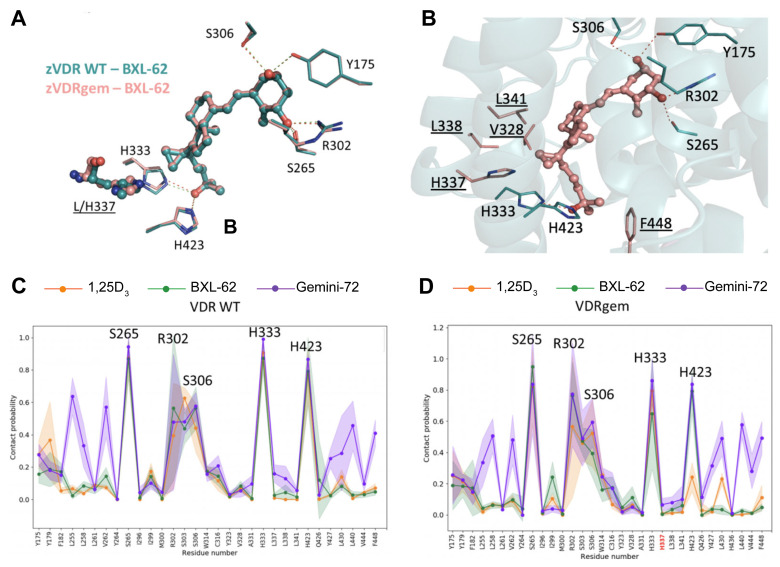
Crystal structure of zVDRgem—BXL-62, and dynamic network of ligand–amino acid contacts in VDR complexes. (**A**) Superposition of BXL-62 on the crystal structures of the zVDR WT (PDB 7BNS) and zVDRgem (PDB 7BNU) complexes; (**B**) Labeled zVDRgem residues correspond to those forming hydrogen bonds with BXL-62 (red) and those underlined correspond to residues that form stronger contacts with BXL-62 than with 1,25D_3_; (**C**,**D**). Contact probabilities between 1,25D_3_ (in orange), BXL-62 (in green) or Gemini-72 (in purple) and zVDR WT (**C**) and zVDRgem (**D**). Five independent 100 ns molecular dynamics (MD) simulations were computed and averaged, and the MD trajectories at the residue level interactions were analyzed with a distance threshold of 3.5 Å.

**Figure 5 ijms-23-08445-f005:**
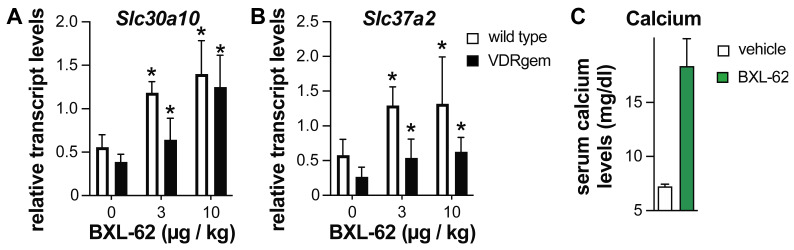
Effect of BXL-62 on intestinal VDR target gene transcript levels and serum calcium levels. Relative duodenal transcript levels of *Slc37a2* (**A**) and *Slc30a10* (**B**) in wild type (white bar) and VDRgem (black bar) mice after a 6-h oral administration of vehicle, 3, and 10 µg/kg of BXL-62. The transcript levels of the 18S housekeeping gene were used as internal control; (**C**) Serum calcium levels of VDRgem mice 24-h after 4 administrations of vehicle (oil), or 10 µg/kg BXL-62 on 4 consecutive days. * *p* < 0.05 vs. vehicle. *n* > 3 mice/group.

**Table 1 ijms-23-08445-t001:** Distance probabilities in zVDR and zVDRgem complexes. The distance criteria used implies that at least one heavy atom from each side chain lies within 3.5 Å of distance.

		His333-Lig	His423-Lig	His333-His423	His423-His337
zVDR WT	1,25D_3_	0.91	0.77	0.17	0
	BXL-62	0.87	0.79	0.15	0
zVDRgem	1,25D_3_	0.80	0.24	0.06	0.53
	BXL-62	0.65	0.79	0.21	0.07

## Data Availability

Atomic coordinates for the X-ray structures of zVDR LBD-BXL-62 (PDB 7BNS) and zVDRgem LBD-BXL-62 (PDB 7BNU) are available at the RCSB Protein Data Bank.

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
