# Peer review of "Vitamin D Analogs Bearing C-20 Modifications Stabilize the Agonistic Conformation of Non-Responsive Vitamin D Receptor Variants"

_ijms, 2022, doi:10.3390/ijms23158445_

Round 1

Reviewer 1 Report

The authors have shown the use of vitamin d receptor stabilization with vitamin D analogue. Tin addition to understanding VDR dependent mechanism, the study has implications for therapy where vitamin d receptor polymorphisms can have detrimental effect. Some minor comments are as below:

For Figure 1, how is the relative activity calculated? Which sample was considered 100%? Did the authors verify the levels of VDR for each treatment conditions? Similar comments for other figures. it’s not very apparent which one is used as control here. Figure 2 “HVDRR: should it be hVDR? How was qPCR relative expression calculated? The authors could include the details of the vdrgem mice. Was vdrgem only expression in the intestines?

Reviewer 2 Report

In this manuscript, Belorusova and collaborators provide insight into the mechanism of action of vitamin D analogs modified in C-20 moiety to have a stronger stimulatory effect on the VDR than natural ligand 1,25D3. This is an exceptional manuscript well-written in which data are strongly supported by a combination of methodological techniques such as crystallography and molecular dynamics analysis.

The authors show that the shown vitamin D analogs on C-20, such as BXL-62, improve transcriptional activity through closer contact with His305 (zHis333) and His397 (zHis423), the addition of 16-ene modification increases the binding affinity for VDR LBP. In both cases, a mutual adaptation of the 16-ene and C-20 moiety and the VDR leads to the formation of an alternative and additional contacts to those present for 1,25D3.

Based on these data, a stronger interaction of ligands with the VDR LBP is suggested to happen and this should be reflected by a decrease in the binding constant to the VDR. The authors are encouraged to report additional experimental data or incorporate some in silico data supporting this point.

Minor corrections.

-          Please normalize the font type and size in all the text. See lines 27-45; 213-228; 335-340
